# Bioactive Metabolite Survey of Actinobacteria Showing Plant Growth Promoting Traits to Develop Novel Biofertilizers

**DOI:** 10.3390/metabo13030374

**Published:** 2023-03-02

**Authors:** Teresa Faddetta, Giulia Polito, Loredana Abbate, Pasquale Alibrandi, Marcello Zerbo, Ciro Caldiero, Chiara Reina, Guglielmo Puccio, Edoardo Vaccaro, Maria Rosa Abenavoli, Vincenzo Cavalieri, Francesco Mercati, Antonio Palumbo Piccionello, Giuseppe Gallo

**Affiliations:** 1Dipartimento STEBICEF, Università degli Studi di Palermo, Viale delle Scienze, 90128 Palermo, Italy; 2Consiglio Nazionale delle Ricerche, Institute of Biosciences and Bioresources (IBBR), Corso Calatafimi 414, 90129 Palermo, Italy; 3Mugavero Teresa S.A.S., Corso Umberto e Margherita n. 1/B, Termini Imerese, 90018 Palermo, Italy; 4Dipartimento AGRARIA, Università Mediterranea di Reggio Calabria, località Feo di Vito, 89122 Reggio, Italy; 5Istituto di Candiolo–Fondazione del Piemonte per l’Oncologia—IRCCS Strada Provinciale, 142-KM 3.95, 10060 Candiolo, Italy; 6National Biodiversity Future Center, Piazza Marina 61, 90133 Palermo, Italy

**Keywords:** PGP traits, bioactive metabolites, VOCs, actinobacteria, streptomycetes, tomato seeds and seedlings, biofertilizers

## Abstract

The use of chemical fertilizers and pesticides has caused harmful impacts on the environment with the increase in economic burden. Biofertilizers are biological products containing living microorganisms capable of improving plant growth through eco-friendly mechanisms. In this work, three actinobacterial strains *Streptomyces violaceoruber*, *Streptomyces coelicolor*, and *Kocuria rhizophila* were characterized for multiple plant growth promoting (PGP) traits such as indole acetic acid production, phosphate solubilization, N_2_-fixation, and drought and salt tolerance. Then, these strains were investigated for their secreted and cellular metabolome, revealing a rich arsenal of bioactive molecules, including antibiotics and siderophores, with *S. violaceoruber* being the most prolific strain. Furthermore, the in vivo assays, performed on tomato (*Solanum lycopersicum* L.), resulted in an improved germination index and the growth of seedlings from seeds treated with PGP actinobacteria, with a particular focus on *S. violaceoruber* cultures. In particular, this last strain, producing volatile organic compounds having antimicrobial activity, was able to modulate volatilome and exert control on the global DNA methylation of tomato seedlings. Thus, these results, confirming the efficacy of the selected actinobacteria strains in promoting plant growth and development by producing volatile and non-volatile bioactive molecules, can promote eco-friendly alternatives in sustainable agriculture.

## 1. Introduction

The use of chemical fertilizers and pesticides has been a strategy to boost agricultural production and meet global food demand for the increasing world population [1]. However, their excessive utilization has caused harmful impacts on the environment, animals and human health, and economic costs [1]. Their negative effects are also exacerbated by the climate change scenario, which increases the multiple and concurrent abiotic stresses (i.e., drought, salinity, low and high temperature, waterlogging, metal toxicity, etc.) representing an additional serious threat to crop yield and food security [2,3]. Therefore, the search for eco-friendly alternatives is a crucial challenge in sustainable agriculture.

A promising strategy would be the use of biofertilizers, which are biological products containing living microorganisms, typically bacteria and fungi, capable of improving plant growth and yield through eco-friendly mechanisms [4]. Biofertilizers, usually applied to soil, seeds, and the plant surface, are often capable of colonizing the rhizosphere and even migrating into the interior part of the plant tissue increasing plant growth and protecting them from abiotic and biotic stress [5]. The mechanisms of microbial stimulation on plant growth can be either direct or indirect, involving the regulation of hormonal, nutritional, and water balance, the solubilization of organic and inorganic phosphates, nitrogen fixation, protection from pathogens by the induction of plant systemic resistance, and/or through the microbial production of antibiotic compounds or enzymes [6,7]. Therefore, biofertilizers are generally formulated using plant growth promoting (PGP) bacteria, which are deemed to mostly contribute to increased crop yields and soil fertility. The ability to exert plant growth promotion seems to be widespread among different kinds of bacteria genera including *Azospirillum*, *Azotobacter*, *Bacillus*, *Enterobacter*, *Pseudomonas*, *Kocuria*, *Klebsiella*, *Streptomyces*, and *Rhizobium* [8,9,10]. Although many aspects of biofertilizer utilization have to be elucidated, such as the plant-specific dosages and long-term effects on microbial biodiversity, currently, the biofertilizer market is greatly increasing, and it is expected to reach USD 3.8 billion by 2025 [7,11]. Thus, the formulation of commercial biofertilizers containing the most effective PGP bacterial strains is desirable. In particular, soil bacteria belonging to the actinobacteria are considered very promising due to their metabolic versatility, drought resistance, capability of supplying nutrients (such as phosphate) for improving growth, and potential to produce a vast array of bioactive metabolites—such as vitamins, antibiotics, plant growth factors—and enzymes [12]. Among these bacteria, *Streptomyces* species are the most prolific source of bioactive metabolites capable of improving plant development and growth and biotic and abiotic stress resistance [8,9,12]. For example, the *Kocuria* genus is gaining a lot of consideration as a PGP bacterium due to its metabolic versatility, resistance to harsh conditions, and production of indole-3-acetic acid (IAA) [13].

In this collaborative research project, involving an academic–industrial consortium and aiming at the development of innovative biofertilizers, three actinobacteria, namely *Streptomyces coelicolor*, *Streptomyces violaceoruber*, and *Kocuria rhizophila*, were selected for possessing at least one PGP trait or for being related to a bacterial genus showing PGP properties, according to the scientific literature. In order to verify their possible utilization for the development of actinobacteria-based biofertilizers, these microorganisms were preliminary characterized for multiple PGP traits, such as IAA, organic and inorganic phosphate solubilization, N_2_-fixation, and drought and salt tolerance. Then, they were analyzed for the secreted and cellular metabolome in axenic cultivations and actinobacterial co-cultures to identify possible bioactive compounds exerting stimulatory effects on plant growth according to the scientific literature. Finally, to evaluate their in vivo PGP capability, tomato (*Solanum lycopersicum* L.) plants were used as the model of study. In addition, global DNA methylation levels and volatilome [14] of the model plant, treated with *S. violaceoruber*, were assayed in comparison with the untreated plants to establish the possible pleiotropic effects on plant physiology.

## 2. Materials and Methods

### 2.1. Bacterial Strains and Culturing Conditions

Actinobacterial strains were used for this study. *Streptomyces coelicolor* M145 and *Kocuria rhizophila* were from our laboratory collection; *Streptomyces violaceoruber* DSM 40783 was purchased from DSMZ (German Collection of Microorganisms and Cell Cultures GmbH). The spore production of *Streptomyces* strains was obtained as previously described [15]. In particular, cultures were performed on mannitol soy flour (MSF) medium at 30 °C for 5–7 days. The spore suspensions were prepared by adding sterile distilled water to the plates. The mixtures were filtered through a syringe containing hydrophilic cotton to eliminate the mycelia. The cell biomass of *K. rhizophila* was obtained in cultures obtained from a single colony inoculated into tryptone soy broth (TSB) and incubated for 24–48 h at 30 °C and 180 rpm. Spore and cell concentrations were evaluated using the colony-forming unit (CFU) method on a tryptone soy agar (TSA) medium. For a long storage time, each bacterial strain was stored at −80 °C in a 20% glycerol solution.

### 2.2. Estimation of PGP Traits and Abiotic Stress Tolerance

The actinobacterial strains were studied for different PGP traits and for tolerance to abiotic stresses (drought and salt resistance/tolerance), phosphate solubilization, indole acetic acid (IAA) production, and nitrogen fixation. All the assays were conducted in triplicate.

#### 2.2.1. Indolic Compound Production

The IAA production by the selected strains was estimated with a colorimetric assay using Salkowski’s reagent (0.5 M FeCl_3_ in 35% HClO_4_ aqueous solution) that is able to reveal in the presence of indole compounds such as IAA [16]. In particular, the strains were grown in 10 mL of R5A [17] and incubated for 24, 48, and 72 h (30 °C, 180 rpm). After incubation, the cultures were centrifuged, and the supernatants were mixed with Salkowski’s reagent (1:1). The optical density (OD) was recorded at 530 nm after 30 min of incubation, and a standard curve with known concentrations (0.5–100 μg mL^−1^) of IAA (Sigma-Aldrich) was used to determine the amount of IAA produced.

#### 2.2.2. Organic and Inorganic Phosphate Solubilization

The ability to solubilize organic and inorganic phosphates was investigated as previously described [18]. In particular, the strains were plated on NBRIP agar media [19] containing different sources of phosphate: AlPO_4_ (5 g L^−1^); Ca_3_P_2_O_8_ (5 g L^−1^); FePO_4_ (5 g L^−1^) to characterize their ability to solubilize inorganic phosphates; and phytate (2 g L^−1^) to characterize their ability to solubilize organic phosphates. In particular, 5 μL of microbial suspension (10^7^–10^8^ CFU mL^−1^) were plated on NBRIP agar media and incubated at 30 °C for 5 days to check the development of a solubilization halo around the colonies. To highlight phosphate solubilization using clearance halo formation, the growth media were provided with bromophenol blue (0.05 g L^−1^), with the exception Ca_3_P_2_O_8_-containing growth medium where the bromophenol blue addition was unnecessary, so that the halos would be visible [19].

#### 2.2.3. Growth under Drought and Salt Stress

All strains were tested for drought resistance and salt tolerance by adding 5% and 15% polyethylene glycol 8000 (PEG) or 7.5% NaCl to the R5A medium, respectively. The cultures were incubated at 30 °C for 48 and 72 h at 180 rpm. The growth was determined by measuring the optical density at 600 nm and then compared with that of the strains incubated in the same conditions but with NaCl or PEG addition.

#### 2.2.4. Growth in Nitrogen-Free Medium

For the evaluation of the nitrogen fixation activity, the bacteria were inoculated into Derxia medium, i.e., a nitrogen-free mineral medium used as an element of discrimination for the cultivated strains [20]. To carry out this test, an amount of 10^7^–10^8^ CFU (i.e., cells and spores from *K. rhizophila* and streptomycetes, respectively), taken from a 20% glycerol solution in a rich growth medium, was inoculated into the test tubes containing 2 mL of Derxia medium. At the same time, with the same procedure described above, cultures were set up in 2 mL of Derxia medium supplemented with the addition of 2 g L^−1^ (NH_4_)_2_SO_4_ as an inorganic source of nitrogen (growth positive control) For the evaluation of the growth of the strains, two parameters were taken into consideration including the culture turbidity and pH variation (given by the bromothymol blue present in the medium).

### 2.3. Metabolite Extraction and HPLC/MS/Q-TOF Analysis

Axenic cultivations or actinobacterial co-cultivations of *S. coelicolor*, *S. violaceoruber,* and *K. rhizophila* were performed by inoculating a suspension of spores (in the case of the two streptomycetes strains) or cells (for *K. rhizophila*) at a concentration of 10^7^–10^8^ CFU mL^−1^ in 50 mL flasks containing 10 mL of R5A medium. The cultures were incubated at 30 °C at 180 rpm for 72 h. Then, 1 mL of cultivations was collected and centrifuged (12,000× *g*, 5 min, 4 °C) to separate the bacterial cells and the spent media that were stored at −20 °C until use. 

HPLC/MS analysis was performed by adapting previously reported methods [21,22]. Samples for HPLC (Agilent 1260 Infinity) were prepared by collecting spent media from cultures (1 mL). The liquid was filtered, diluted with 1 mL of MeOH, and directly injected. Analyses were performed in triplicate. Water and acetonitrile were of the HPLC/MS grade. Formic acid was of analytical quality. A reversed-phase Phenomenex Luna C18(2) column (150 mm × 4.6 mm, particle size 3 µm) with a Phenomenex C18 security guard column (4 × 3 mm) was used. The injection volume was 25 µL. The eluate was monitored with Mass Total Ion Count (MS TIC) and UV (270 nm). Mass spectra were obtained on an Agilent 6540 UHD accurate-mass Quadrupole-Time of flight (Q-TOF) spectrometer equipped with a Dual AJS Electrospray Ionization (ESI) source working in positive or negative mode. Nitrogen N_2_ was used as desolvation gas at 300 °C and a flow rate of 8 L min^−1^. The nebulizer was set to 45 psig. The sheath gas temperature was set at 400 °C and a flow of 12 L min^−1^. A potential of 2.6 kV and 3.2 kV was used on the capillary for the negative and positive ion modes, respectively. The fragmentor was set to 75 V. MS spectra were recorded in the 150–1000 m/z range. Quality control was performed prior to analysis using mass calibration in the range of 100–3000 Dalton (Q-TOF calibration mix) and solvent delay calibration for retention time. An in-house quality check mix containing known compounds (phenylalanine, saccharose, benzoic acid, and rutin) was injected during the batch of analysis. Mass spectrum data were analyzed for metabolites annotation using MassHunter Qualitative Analysis B.06.00 and the Metlin database [23]. Statistical significance of the quantitative data for each metabolite was calculated using one-way ANOVA with *p* < 0.05 [24].

### 2.4. In Vivo Evaluation of Plant Growth Promotion by PGP Actinobacteria

Seeds of *Solanum lycopersicum*, L., a commercial tomato variety (Red Cherry), were surface-sterilized with 70% (*v*/*v*) ethanol for 1 min, followed by treatment with 2.5% (*w*/*v*) sodium hypochlorite solution for 2 min, as previously reported [18]. Afterward, three rinses in sterile distilled water (5 min each) were carried out. Surface-sterilized seeds were immersed for 45 min into a dilution (1:10 in distilled water) of PGP actinobacterial cultivations (about 3% *v/v* packed mycelium and 10^9^–10^10^ CFU mL^−1^ for the two streptomycetes and *K. rhizophila*, respectively) performed in R5A growth medium at 30 °C for 72 h. In parallel, control condition experiments were performed using sterile distilled water instead of diluted PGP actinobacterial cultures. Five replicates, which included ten seeds each, were aseptically transferred to Petri dishes with filter paper (Whatman) soaked in 3 mL of distilled sterile water. Plates were incubated in dark at 25–26 °C for six days for germination and after they were moved in a growth-controlled chamber (25 ± 2 °C, 65–75% relative humidity, 16 h of daylight with a light intensity of 3000 lux) for other six days. The plant responses (root and shoot growth at 12 days after treatment—DAT) to each thesis were analyzed with the one-way ANOVA test using R-packages [25], considering at least 3 replicates (30 plants) for each treatment. Tukey’s test (*p* < 0.05) was used to test the significance among different treatments [25].

### 2.5. Effect of S. violaceoruber Culture Seed-Priming Treatment on Germination

For this experiment, seeds of *Solanum lycopersicum*, L., commercial variety UC82, were sterilized as above described. Finally, seeds were washed with distilled water to remove residual ethanol and NaOCl. After sterilization, seeds were treated with three different conditions: Milli-Q water (control condition) and 1:5 (T1) and 1:10 (T2) dilute water solution of *S. violaceoruber* cultures (about 3% *v/v* packed mycelium) performed in R5A growth medium at 30 °C for 72 h. The treatment was applied drop by drop to dried seeds shaking until the complete and visible distribution of the product on the seed surface was obtained [26]. Following the treatment, seeds were dried at room temperature and then placed in Petri dishes (9 cm Ø) (ten seeds per plate) containing two filter papers saturated with 3.5 mL distilled water. Five replicates were performed for each treatment. The Petri dishes were then placed in the dark in a growth chamber at 24 °C temperature, 65% relative humidity, and a photoperiod of 14 h with 350 μmol light intensity for 7 days. After this period, the germination percentage, germination index (GI %), and root and hypocotyl length were evaluated as reported above:$$\%\;\mathrm{Germination}=\frac{Germinated\;seeds\;number}{total\;seed\;number}\cdot 100$$
$$\mathrm{Germination}\;\mathrm{Index}\;(\mathrm{GI}\;\%)=\frac{average\;number\;of\;Germinated\;seeds\;\left(T\right)*average\;L_{R}\left(T\right)}{average\;number\;of\;Germinated\;seeds\;\left(C\right)*average\;L_{R}\left(C\right)}*100$$

*L_R_* = Root length;

(*T*) = Treated;

(*C*) = Control.

Morphological measurements of root and hypocotyl length were made by placing the seedlings on graph paper and analyzed with ImageJ software.

### 2.6. Identification of VOCs Using SPME-GC/MS

GC/MS analysis was performed by adapting previously reported methods [27]. In particular, vials for solid-phase microextraction (SPME) were partially filled either with (i) *S. violaceoruber* mycelial biomass on the excised surface of 2 g R5A-agar growth medium plugs obtained after 3 days of incubation at 30 °C or (ii) with freshly collected *S. lycopersicum* seedlings, regenerated on filter paper in Petri dishes and either treated with *S. violaceoruber* cultures or untreated (control condition) as above described. Then, volatile organic compounds (VOCs) were extracted from the sealed vial headspace and concentrated using SPME before desorption in the GC injection port. Headspace extraction was performed with a 2.5 mL Syringe-HS (0.64-57-R-H, PTFE, GERSTEL) conditioned and held at 40 °C from sample collection to injection.

In the SPME, one Fiber Assembly was evaluated and used: 50/30 µm divinylbenzene (DVB)/carbowax (CAR)/polydimethylsiloxane (PDMS) (Supelco, Bellefonte, PA, USA). The fiber was exposed to bacterial culture in a 20 mL SPME vial (75.5 × 22.5 mm) for 30 min at 40 °C after 30 min of equilibration time. The desorption time was 5 min. Before use, the fiber was conditioned and cleaned at 270 °C for 30 min, following the instructions from Supelco^®^. Splitless injection was used.

Gas chromatographic analysis was performed using an Agilent 7000C GC (Agilent Technologies, Inc., Santa Clara, CA, USA) system equipped with a split/splitless injector, fitted with an Agilent HP5-MS UI capillary column (30 m × 250 μm; 0.25 μm film thickness) coupled to an Agilent triple quadrupole Mass Selective Detector MSD 5973 (Agilent Technologies, Inc., Santa Clara, CA, USA), with ionization voltage of 70 eV; electron multiplier energy of 2000 V; and transfer line temperature of 270 °C. The solvent Delay was 0 min. Helium was used as the carrier gas (1 mL min^−1^). The oven program was as follows: the temperature was initially kept at 40 °C for 5 min and then gradually increased to 250 °C at a rate of 2 °C/min, which was held for 15 min and finally raised to 270 °C at 10 °C/min. Samples were injected at 250 °C automatically. The interval scan was 35–450 *m*/*z* and the scan speed was 10,000 amu·s^−1^ (25 Hz).

The GC–MS mass spectrum data were analyzed using MassHunter Qualitative Analysis B.06.00, and the database of the National Institute Standard and Technology (NIST) was used to interpret the analyzed data. A comparison of the mass spectrum of the unidentified components released by the bacterial isolates was carried out against the mass spectrum of already-known components available in the NIST 11 MS library. Statistical significance of the quantitative data for each metabolite was calculated using one-way ANOVA with *p* < 0.05 [24].

### 2.7. Quantification of Global DNA Methylation

A total of nine individual *S. lycopersicum* seedlings, regenerated on filter paper in Petri dishes and either treated with *S. violaceoruber* cultures or untreated as above described, were hand dissected. Then, shoots were pooled into three independent biological samples. The DNA isolation was carried out with the Plant Genomic DNA Purification Kit (Macherey-Nagel).

Global DNA methylation levels, referred to as the total level of 5-methylcytosine (5-meC) content in a sample, were quantified using 100 ng of genomic DNA from each sample and a MethylFlash Methylated DNA Quantification Kit (Epigentek), according to the manufacturer’s instructions. Briefly, 100 ng of genomic DNA samples were bound to an ELISA plate and fluorescently labeled for 5-methyl Cytosine (5-meC) presence using specific antibodies. Each sample was run in duplicates along with internal controls provided by the kit, and the optical density (OD) intensity was measured for the plate based on the amount of 5 meC absorbance at 450 nm. The slope of the standard curve generated by positive controls was determined using linear regression and used to identify the global 5 meC amount of each sample. The percentage of global DNA methylation was then calculated as a ratio of its OD relative to the OD of positive controls, after subtracting the negative control OD values. Data are presented as a mean ± standard error (SE). Statistical significance of quantitative data was calculated using one-way ANOVA with *p* < 0.05 [24].

## 3. Results

### 3.1. Looking for Multiple PGP Traits of Three Selected Actinobacteria

As preliminary characterization, the actinobacteria strains *S. coelicolor*, *S. violaceoruber*, and *K. rhizophila* showed to be positive for all (*S. violaceoruber*) or most (*S. coelicolor* and *K. rhizophila*) of the assayed PGP traits including (i) producing IAA, (ii) solubilizing organic and inorganic phosphate salts, (iii) using atmospheric nitrogen to grow, and (iv) growing under saline and drought stresses (Table 1; Appendix A). 

These results confirm and expand those obtained in previous works concerning some PGP traits of the same or closely related strains, thus highlighting the potential application of actinobacteria to develop novel biofertilizers due to their different metabolic capabilities, which can directly or indirectly improve plant growth, nutrient acquisition, and abiotic stress tolerance of plants [10,28,29,30].

### 3.2. Bacterial Metabolomic Analyses of Single and Mixed Actinobacterial Cultures

*K. rhizophila, S. coelicolor*, and *S. violaceoruber* were cultivated in R5A growth medium as axenic- or co-cultivation to analyze the bacterial secreted and cellular metabolome. *S. coelicolor*, included in this assay, is a model strain for streptomycetes as its genome has been sequenced [31] and its secretome has been characterized [32,33]. Metabolites from liquid cultures were identified using HPLC/ESI/MS/Q-TOF. In particular, compounds released into the supernatant were directly analyzed after culture centrifugation, while intracellular metabolites were extracted from the pellet with methanol. A total of 33 metabolites was identified and quantified from 7 different experiments including axenic cultures, double and triple co-cultivations, and spent medium mixes (all data are reported in Table 2). Figure 1 reports most of the secreted metabolites with quantitative variations. The identified compounds belong to different classes of bacterial metabolites, among these various compound were annotated related to tryptophan metabolism (3-methyl-indole, L-tryptophan, kynurenic acid, *N*-acetyl-L-tryptophan), various polyketides related to the biogenesis of actinorhodin (frenolicin E, frenolicin E isomer, nanaomycin E, nanaomycin A, nanaomycin E isomer, nanaomycin A isomer, DMAC, aloesaponarin, actinorhodin), four alkaloids (coelimycin P1, carbazomycin F, pimprinethine, streptazolin), two siderophores (desferrioxamine B, nocardamin), one inosine (futalosine), one diterpenoid (anthranoyllycoctonine), two phenezines alkaloids (streptophenazine F, streptophenazine A), two sesquiterpene lactone (arenaemycin E, pentalenolactone E), one isoflavone (3′,8-Dihydroxy-4′,6,7-trimethoxyisoflavone), three pyranone polyketides (Germicidin A, B, and D), one antibiotic polypeptide (Calcium-dependent antibiotic CDA4b), one Manumycin (Colabomycin E), and two prodiginine antibiotics (streptorubin B, undecylprodigiosin) (Table 2). It is interesting to note that the most prolific strain in terms of the number of produced metabolites is *S. violaceoruber* and, in some cases, co-cultivations determine stimulation of metabolite production, such as in the case of colabomycin E, carbazomycin F and arenaemycin E, and in other cases, a decrease in a metabolite-specific manner (Figure 1; Table 2). Overall, *Streptomyces*–*Kocuria* co-cultivations resulted in a similar production or a production decrement in the respect of axenic cultures.

### 3.3. Biostimulant Effects on S. lycopersicum Seedlings from PGP Actinobacteria Treated Seeds

Overall, the seed treatments performed using actinobacteria cultivations resulted in a general trend of an increased root and shoot length in the regenerated seedlings in comparison to the control condition (Figure 2 and Appendix A). In addition, the effect of the PGP actinobacteria slightly differed on root and shoot length, and not all the increments were significant. Indeed, although in the shoot, a general increase was observed for all PGP actinobacterial cultivations, showing an increment in the total shoot length ranging from 5.05% to 43.44% for the *S. violaceoruber* and *K. rhizophila* culture mixture (VIOL + KOC) and *S. coelicolor* cultures (M145), respectively, significant values were recorded for *S. violaceoruber*, *S. coelicolor*, and *K. rhizophila* single culture treatments (VIOL, M145, and KOC, respectively) (Figure 2; Table 3). Among the mixtures, only *S. violaceoruber* and *S. coelicolor* culture mixture (VIOL + M145) had a significant increase in the shoot length (Figure 2). In agreement, all PGP actinobacteria culture treatments displayed a clear growth promoting effect on the root, with a growth rate from 38.03% to 78.24% (Table 3). All treatments had significantly increased values in comparison to the control (Figure 2; Table 3), with a higher effect on the root length for M145 and VIOL + M145 that showed roots nearly two times greater than untreated plants.

### 3.4. Effect of PGP Seed-Priming Treatment on Germination

The *S. violaceoruber* culture seed-priming treatment, at both concentrations, did not significantly affect the tomato seed germination, which did not differ from the control values. Conversely, at the highest concentration, the *S. violaceoruber* culture seed-priming treatment significantly increased the seed germination index % and root and hypocotyl length compared to the control seeds (Table 4).

### 3.5. Volatile Organic Compounds Produced by S. violaceoruber and S. lycopersicum

In order to obtain a deeper picture of the biological mechanisms involved in the bacterium–plant interaction, VOCs produced by *S. violaceoruber* and *S. lycopersicum* plants treated or not treated with *S. violaceoruber* were identified using SPME GC/MS. Eight bacterial metabolites were revealed and are reported in Table 5. Among the identified compounds, there are some of particular interest such as sulfur-containing compounds, antibiotic dimetildisulfide or dimethyltrisulfide [34,35], and characteristic terpenes produced by streptomycetes, such as 2-methylisoborneol (MIB) and geosmin [36]. Concerning *S. lycopersicum* metabolites, all nine identified compounds were common monoterpenes produced by tomato plants (Table 6) [37]. The quantitative evaluation of tomato VOCs revealed a reduction in all the identified compounds upon the treatments (Figure 3).

### 3.6. Effect of S. violaceoruber Cultivation on Global DNA Methylation Amount of S. lycopersicum Shoots

Global DNA methylation analysis in the shoots from tomato plants grown on filter paper in Petri dishes revealed that PGP treatment elicited hypermethylation (Figure 4). In particular, the averaged 5 meC percentages increased from 2.1 ± 0.74 to 2.73 ± 0.17 following PGP exposure. It is worth mentioning, compared to the sibling control untreated plants, the shoots from PGP-treated plants exhibited quite similar methylation levels to each other, suggesting that PGP exposure elicited higher inter-individual uniformity in global 5 meC amounts. 

## 4. Discussion

The characterization of PGP bacteria that positively influence plant growth and development has been widely discussed in recent works to develop strategies for ecosystem friendly and sustainable agriculture. The PGP bacteria can promote plant growth directly—involving the production of phytohormones, facilitating the uptake of nutrients such as phosphorus, from the environment, mitigating drought and saline stress—or indirectly—by preventing the deleterious effects of phytopathogens that produce bioactive metabolites. Many studies have been conducted on actinobacteria, highlighting the ability of these microorganisms to promote plant growth and their synergistic effects on plant growth and protection [38].

The actinobacteria usually are soil inhabitants where they have a saprophytic lifestyle, and they are often isolated from the rhizosphere and root tissues [29,39]. Rhizosphere actinobacteria represent a major component of rhizosphere microbial populations, with economic importance for humans. In fact, the productivity of agricultural and forest fields depends on their contributions to soil systems [40], significantly influencing nutrient cycling, and improving plant health and growth. In this study, we characterized the three actinobacteria *S. coelicolor*, *S. violaceoruber,* and *K. rhizophila*. As it has been previously described [41], they can be considered PGP rhizobacteria since they satisfy at least two of the following three criteria: plant colonization, plant growth stimulation, and biocontrol. Indeed, the results of this study revealed that as the rhizosphere bacteria possessing different PGP traits—such as bioactive compound production and growth using different inorganic or organic phosphate sources and under drought and saline stress—these actinobacterial strains can enhance plant growth by improving the availability of mineral nutrients through siderophore production, phosphate solubilization, and putative diazotrophic activity; stimulating plant development by phytohormone, antimicrobial, and VOC production; and increasing plant tolerance to drought and salt stresses. The *Streptomyces* tolerance to abiotic stresses is essential not only for the survival of the microorganism itself but also because some streptomyces enhance PGP traits such as the production of phytohormones and siderophores in saline soil conditions, relieving at the same time the plants from stress and, thus, improving its health, growth, and development [42]. In addition, the possible occurrence of the ability to grow in a nitrogen-free medium by *S. violaceoruber* is interesting since it suggests that *S. violaceoruber* is able to grow using atmospheric nitrogen. This result is in agreement with other studies on nitrogen-fixing properties, which revealed this ability in some actinobacteria including *Streptomyces* species [12,43]. Although this point is really crucial and fascinating also for an ecological perspective concerning the importance and spreading of this microbial process, it deserves further analyses such as the nitrogen isotope uptake coupled with MS analysis together with an elucidation of the possible metabolic pathways and enzymes involved in this process. Nevertheless, concerning the development of novel biofertilizers, the eventual contribution of this strain in relieving nitrogen limitation stress on the plant has to be analyzed.

For the in vivo assays, the tomato—which is one of the most important crops in the world and a model plant for the study of growth and fruit development—was used [44]. The assay was performed using surface sterilized seeds regenerated in filter papers in Petri dishes after treatment using different bacterial cultivations or cultivation mixtures. The PGP actinobacteria were cultivated in the R5A growth medium that is suitable for stimulating bioactive metabolite production in streptomycetes [17] and, at the same time, it does not have any stimulatory effect on *S. lycopersicum* seed germination and/or growth and development according to our preliminary investigations. In general, the treatment using bacterial cultures provided for an improvement in rooting development, as compared to the untreated control. In addition, tomato seed germination was not improved by the *S. violaceoruber* culture priming treatment. Similar results were found in tomato seeds treated with mugwort aqueous extract [45]. However, the increase in the IG % and the root and hypocotyl length underlined the positive effect of PGP-priming treatment on seedling growth. This effect may be due to the high capacity of this strain to produce IAA, an important growth promoter [46]. Therefore, the *S. violaceoruber* culture seed-priming treatment is not harmful to seeds and has not shown phytotoxicity as biostimulants used in other studies [47]. Furthermore, it could be used to enhance the initial tomato seedling growth, in both open fields and protected cultivation, also increasing its performance in the transplanting of seedlings in a horticultural nursery.

The use of actinobacteria, with *Streptomyces* spp. being the most studied particularly as biocontrol agents to contrast various bacteria and fungi causing plant diseases [48,49,50,51,52,53], has been described. These studies highlighted the production of an arsenal of different bacterial volatile and non-volatile bioactive molecules that can promote plant growth by increasing tolerance against abiotic and biotic stresses and by stimulating plant development as inferred from the scientific literature describing their action mechanisms [51,54,55]. For example, the fermentation broth of *Streptomyces* sp. AN090126 has been demonstrated to exhibit antimicrobial activity against bacterial and fungal pathogens. In this context regarding biological activities of PGP compounds, this study highlighted the production of different kinds of interesting bioactive molecules such as germicidins (antibiotics, spore germination regulators) [56]; siderophores such as desferrioxamine E (iron uptake and transport) [57] and nocardamin (prevents plant defense against infections) [58]; many antibiotics such as CDA, prodiginines, and actinorhodin [59]; and tryptophan metabolites are related to IAA (PGP compound) [60], while kynurenic acid play a role on ethylene/auxin balance [61].

The results concerning the quantitative variations in tomato VOCs due to PGP bacterial treatment are interesting since they highlight an interaction between actinobacteria and plants which may control different aspects of plant physiology, thus suggesting a pleiotropic level of regulation. The identified tomato VOCs are terpenes of which numerous studies have reported pharmacological properties including antioxidant, anti-inflammatory, antiparasitic, antidiabetic, antiviral, antitumor, antibacterial, and antifungal activities [62,63,64]. As an example, the antimicrobial effect of monoterpenoids such as γ-terpinene has been ascribed to a perturbation of the lipid layer of the microbial plasma membrane [65]. A possible interpretation of the reduction in terpene levels in the treated plants could be ascribed to the bacterial strain and plant interaction to avoid any damage to the PGP strain due to the antimicrobial effect of terpenes that can be replaced with the plethora of bioactive compounds produced by the bacterial strain, including the VOCs that were identified considering *S. violaceoruber* alone. Indeed, it has been reported that dimethyl disulfide and dimethyl trisulfide inhibit the growth of *Rhizoctonia solani* and *Pythium ultimum* and many Gram-negative and Gram-positive bacteria [34,35]. In addition, the abundance of dimethyl disulfide positively correlated with the antimicrobial activity of *Arthrobacter* sp. OVS8 against *Burkholderia cepacia* complex strains [27] and VOCs produced by *Streptomyces* sp. AN090126, and including dimethyl sulfide and trimethyl sulfide inhibited the growth of pathogenic bacteria and fungi in vitro [66]. Interestingly, it has been recently shown that the interaction of *Streptomyces. rochei* with *Fusarium moniliforme* and *Curvularia lunata*, two sorghum grain mold pathogens, affects the production of microbial mVOCs, suggesting also a stimulatory role on plant growth [67]. In addition, another study revealed the potential contribution of bacterial endophytes to the production of plant essential oils [27]. Thus, all together, this data suggest that the bacteria may have a key role in the production of terpenes from plants and vice versa and that VOC production is an interesting aspect of plant and prokaryotic interaction and coevolution that still deserves further investigations. In this context, it should be emphasized that plant epigenomes are heavily susceptible to environmental variation, essentially because plants are sessile organisms that must respond to or endure continuous challenges in their surrounding environment [68]. Indeed, the observed change in global DNA methylation elicited by PGP exposure represents a valuable starting point for further studies aimed to define correlations among epigenotype, gene expression, and phenotype after exposure of tomato plants to PGP bacterial strains.

## 5. Conclusions

In this study, three actinobacterial strains *S. violaceoruber*, *S. coelicolor*, and *K. rhizophila* were characterized for multiple PGP traits such as IAA production, organic and inorganic phosphate solubilization, N_2_-fixation, and drought and salt tolerance. Then, these strains were also investigated for their secreted and cellular metabolome, revealing a rich arsenal of bioactive molecules, including antibiotics and siderophores, with *S. violaceoruber* being the most prolific strain. The actinobacterial PGP trait characterization and metabolomic investigations paralleled the in vivo assay on *S. lycopersicum* seeds and seedlings, confirming the efficacy of the selected strains in promoting plant growth and development. In fact, these in vivo assays, performed on tomato (*S. lycopersicum* L.), resulted in an improved germination index and growth of seedlings from seeds treated with PGP actinobacteria, with a particular focus on *S. violaceoruber* cultures. This actinobacterial strain, producing volatile organic compounds having antimicrobial activity, was also able to modulate volatilome and to exert control on the global DNA methylation of tomato seedlings.

Thus, the results of this study suggest the possible use of these three promising actinobacterial strains as biofertilizers that are potentially able to preserve PGP traits even under adverse conditions such as salinity and drought, thus promoting plant growth and increasing tolerance against abiotic and biotic stresses. Detailed investigations of the metabolome, proteome, and VOCs, as well as epigenetic insights on mature plants, will be able to provide insights on novel plant–bacterial interaction mechanisms and will allow the identification of new active biomolecules to be applied in eco-sustainable agriculture, thus reducing the use of pesticides.

## Figures and Tables

**Figure 1 metabolites-13-00374-f001:**
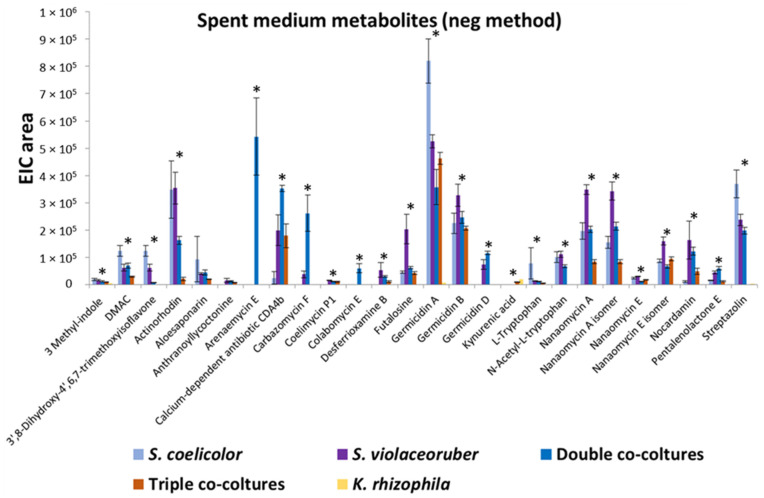
Quantitative profiles of extracellular metabolites reported as extracted ion chromatogram (EIC) area and identified in *K. rhizophila, S. coelicolor,* and *S. violaceoruber* cultivations, in *S. coelicolor* and *S. violaceoruber* (double) co-cultures, and in *S. coelicolor*, *S. violaceoruber,* and *K. rhizophila* (triple) co-cultures. The values are reported as the mean of three cultivations; SE values are also reported as bars. Asterisks indicate quantitative group differences that are statistically significant (*p* < 0.05) according to the one-way ANOVA test.

**Figure 2 metabolites-13-00374-f002:**
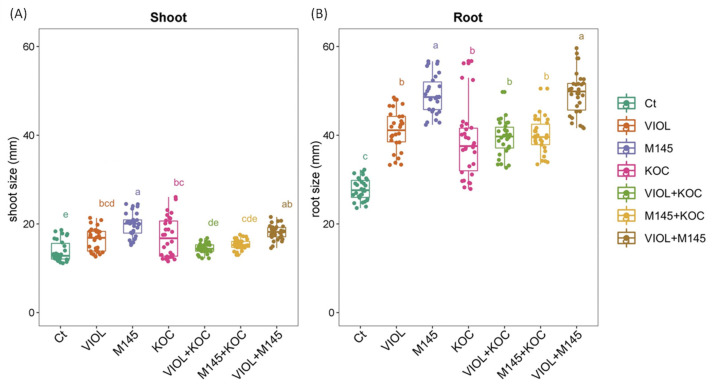
Tomato shoot (**A**) and root (**B**) responses to the PGPB treatments after 12 DAT analyzed with one-way ANOVA followed by Tukey’s post hoc test (*p* < 0.05). Different letters indicate statistically significant differences among treatments in pairwise comparisons. Ct: control; VIOL: *S. violaceoruber*; M145: *S. coelicolor*; KOC: *K. rhizophila*; VIOL + KOC: *S. violaceoruber* and *K. rhizophila* mixes; M145 + KOC: *S. coelicolor* and *K. rhizophila* mixture; VIOL + M145: *S. violaceoruber* and *S. coelicolor* mixture.

**Figure 3 metabolites-13-00374-f003:**
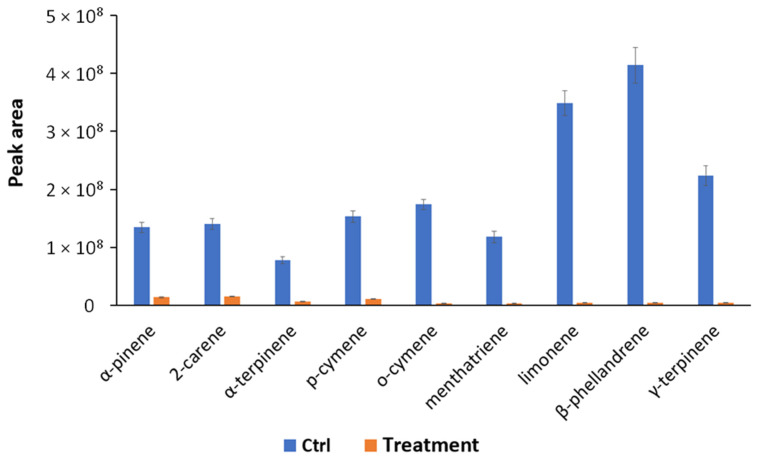
Quantitative profiles of VOCs reported as peak area and produced by *S. lycopersicum* treated (Treatment) and untreated (Ctrl, control condition) identified using SPME-GC/MS. All quantitative group differences are statistically significant (*p* < 0.05) according to the one-way ANOVA test.

**Figure 4 metabolites-13-00374-f004:**
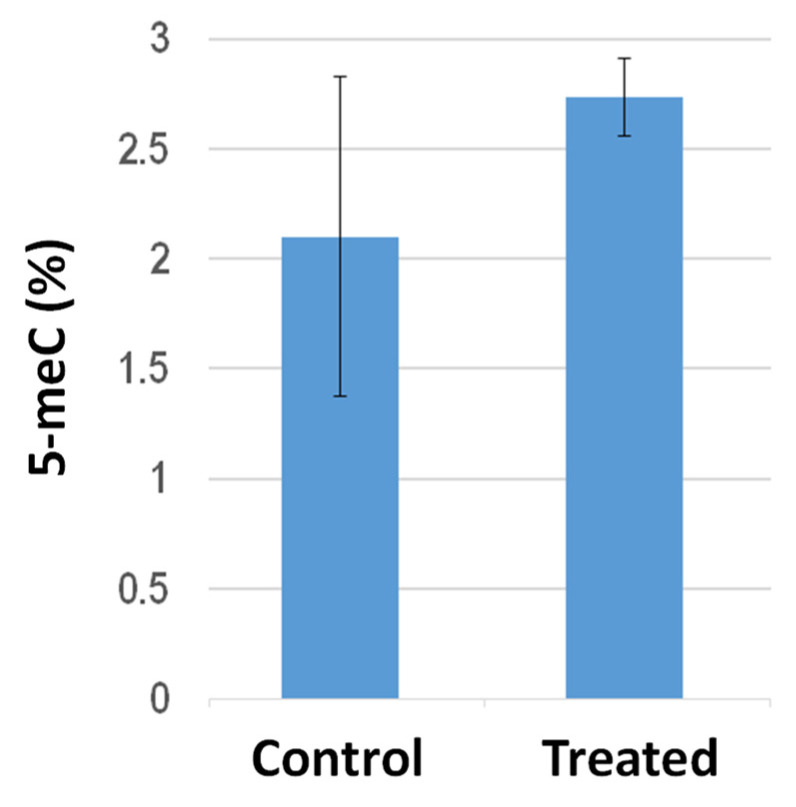
Histogram representing the global 5 meC level measured with ELISA-based assays in shoots from tomato plants either untreated (control) or exposed (treated) to *S. violaceoruber* cultures. Data are presented as a mean ± SE. Quantitative group difference is statistically significant (*p* < 0.05) according to the one-way ANOVA test.

**Table 1 metabolites-13-00374-t001:** Characterization of multiple PGP traits of actinobacteria strains investigated.

Strain	IAA Production	Growth Using FePO_4_	Growth Using Ca_3_P_2_O_8_	Growth Using AlPO_4_	Growth Using Fitate	Growth Using N_2_	Growth under Drought Stress	Growth under Saline Stress
*S. coelicolor*	+	g^+^	g^+^	g^+^	g^+^	N.A.	+	N.D.
*S. violaceoruber*	+	g^+^	g^+^	g^+^	g^+^	+	+	+
*K. rhizophila*	+	g^+^	g^+^	g^+^	g^+^	N.A.	+	+

+, positive result. g, growth. g^+^, growth and solubilization halo. N.A., not applicable since unable of growing in the medium with the addition of (NH_4_)_2_SO_4_ as nitrogen source. N.D., not determined in this study.

**Table 2 metabolites-13-00374-t002:** Metabolites identified using HPLC/MS/ESI/Q-TOF from spent media of bacterial cultivations and their occurrence in different conditions.

t_R_ (min)	Compounds	Molecular Formula	ESI^−^ [M − H]^−^ (*m*/*z*) Exp.	ESI^+^ [M + H]^+^ (*m*/*z*) Exp.	Classes	Occurrence ^1^	
1.22	3 Methyl-indole ^a^	C_9_H_9_N	130.0869	-	Tryptophan metabolism	1–7 neg	-
2.41	L-Tryptophan ^a^	C_11_H_12_N_2_O_2_	203.0822	205.0980	Amino Acid	1–6 neg	1–6 pos
2.69	Kynurenic acid	C_10_H_7_NO_3_	188.0332	190.0509	Tryptophan metabolism	5–7 neg	5–7 pos
3.41	Frenolicin E	C_18_H_20_O_8_	-	365.1190	Polyketide	-	2–6 pos
3.68	Frenolicin E isomer	C_18_H_20_O_8_	-	365.1190	Polyketide	-	2–6 pos
4.20	Coelimycin P1	C_17_H_20_N_2_O_4_S	347.1033	349.1251	Alkaloid	2–5 neg	2–6 pos
4.27	Desferrioxamine B	C_25_H_48_N_6_O_8_	559.3428	561.3649	Siderophore	2–6 neg	2–6 pos
4.62	Carbazomycin F	C_16_H_15_NO_4_	284.0938	-	Alkaloid	2–4, 6 neg	-
4.71	Futalosine	C_19_H_18_N_4_O_7_	413.1090	415.1260	Inosine	1–6 neg	1–6 pos
4.72	Streptophenazine F	C_25_H_30_N_2_O_5_	-	439.2264	Phenazine	-	3,4,6 pos
4.82	Pimprinethine ^a^	C_13_H_12_N_2_O	-	213.1034	Alkaloid		1–7 pos
4.84	Anthranoyllycoctonine	C_32_H_46_N_2_O_8_	585.3142	587.332	Diterpenoid	2–6 neg	2–6 pos
5.07	Nocardamin	C_27_H_48_N_6_O_9_	599.3336	601.3565	Siderophore	1–6 neg	1–6 pos
5.15	Streptazolin ^a^	C_11_H_13_NO_3_	206.0820	-	Alkaloid antibiotic	1–4,6,7 neg	-
5.30	Nanaomycin E	C_16_H_14_O_7_	317.0659	319.0833	Polyketide	1–6 neg	1,2,3,5 pos
5.46	N-Acetyl-L-tryptophan	C_13_H_14_N_2_O_3_	245.0922	-	Tryptophan metabolism	1–4, 6 neg	-
5.47	Nanaomycin A ^a^	C_16_H_14_O_6_	301.0707	303.0880	Polyketide	1–6 neg	1–6 pos
5.48	Nanaomycin E isomer	C_16_H_14_O_8_	317.0658	-	Polyketide	1–6 neg	-
5.56	Streptophenazine A	C_24_H_28_N_2_O_5_	-	425.2081	Phenazine	-	2–6 pos
5.69	Arenaemycin E ^a^	C_15_H_16_O_5_	275.0906	277.1078	Sesquiterpene lactone	3–6 neg	3–6 pos
5.87	3′,8-Dihydroxy-4′,6,7-trimethoxyisoflavone	C_18_H_16_O_7_	343.0700	-	Isoflavone	1–4, 6 neg	-
6.20	Nanaomycin A isomer	C_16_H_14_O_6_	301.0709	303.0880	Polyketide	1–6 neg	1–6 pos
6.40	Germicidin B ^a^	C_10_H_14_O_3_	181.0866	183.1027	Pyranone Polyketide antibiotic	1–6 neg	1–6 pos
6.55	Pentalenolactone E	C_15_H_18_O_4_	261.1118	-	Sesquiterpene lactone	1–6 neg	-
7.15	Germicidin A ^a^	C_11_H_16_O_3_	195.1027	197.1184	Pyranone Polyketide antibiotic	1–7 neg	1–7 pos
7.20	Calcium-dependent antibiotic CDA4b	C_67_H_80_N_14_O_26_	1495.5206	1497.5536	Polypeptide antibiotic	1–6 neg	2–6 pos
7.72	Colabomycin E	C_32_H_32_N_2_O7	555.2174	-	Manumycin	3,4,6 neg	-
7.85	Germicidin D ^a^	C_11_H_16_O_4_	211.0962	-	Pyranone Polyketide antibiotic	2–4,6 neg	-
7.98	3,8-Dihydroxy-1- methylanthraquinone-2-carboxylic acid (DMAC) ^a^	C_16_H_10_O_6_	297.039	-	Anthracene polyketide	1–6 neg	-
9.39	Aloesaponarin ^a^	C_15_H_10_O_4_	253.0501	-	Anthracene (polyketide)	1–6 neg	-
9.56	Actinorhodin ^a^	C_32_H_22_O_14_	629.0917	631.1143	Polyketide antibiotic	1–6 neg	1–6 pos
10.21	Streptorubin B ^a^	C_25_H_33_N_3_O	-	392.2730	Prodiginine antibiotic	-	1–6 pos
11.12	Undecylprodigiosin ^a^	C_25_H_35_N_3_O	392.2692	394.2881	Prodiginine antibiotic	1–6 neg	1–6 pos

^1^ Experiments were referred from 1 to 7 as follows: 1, *S. coelicolor*; 2. *S. violaceoruber*; 3, *S. coelicolor* and *S. violaceoruber* co-cultivations; 4, *S. coelicolor* and *S. violaceoruber* spent medium mix; 5, *S. coelicolor*, *S. violaceoruber*, and *K. rhizophila* co-cultivations; 6, *S. coelicolor*, *S. violaceoruber* and *K. rhizophila* spent medium mix; 7, *K. rhizophila* cultivations. pos and neg refer to the positive or negative mode for ion mode, respectively. ^a^ Compound also present in the extract from cell biomass.

**Table 3 metabolites-13-00374-t003:** Effect on the root and shoot of *S. lycopersicum* seedlings due to PGP actinobacteria treatments on seeds.

Thesis ^1^	Tissue	Mean Value ^2^	Standard Deviation	Q_3_ ^3^	Growth Rate (%) ^4^
Ct	Root	27.81	2.43	29.82	-
	Shoot	13.80	2.38	15.61	-
VIOL	Root	41.16	4.51	44.25	47.97
	Shoot	16.41	2.62	18.32	18.90
M145	Root	49.24	4.35	52.01	77.04
	Shoot	19.79	2.71	20.92	43.44
KOC	Root	38.39	7.97	41.58	38.03
	Shoot	16.92	4.50	20.66	22.61
VIOL + KOC	Root	39.40	3.91	41.81	41.65
	Shoot	14.49	1.18	15.28	5.05
M145 + KOC	Root	39.87	3.88	42.48	43.33
	Shoot	15.36	1.19	16.06	11.35
VIOL + M145	Root	49.58	4.90	51.68	78.24
	Shoot	18.14	1.66	19.22	31.45

^1^ Ct: control; VIOL: *S. violaceoruber*; M145: *S. coelicolor*; KOC: *K. rhizophila*; VIOL + KOC: *S. violaceoruber* and *K. rhizophila* mixes; M145 + KOC: *S. coelicolor* and *K. rhizophila* mixture; VIOL + M145: *S. violaceoruber* and *S. coelicolor* mixture. ^2^ Mean values calculated from 3 biological replicates per thesis, consisting of 10 plants each (in total 30 plants). ^3^ Third quartile (Q3), the middle value between the median and the highest value (maximum) of the data set. It is known as the upper or 75th empirical quartile, as 75% of the data lies below this point. ^4^ Relative growth rate evaluated by the comparison between the mean value of Ct and each PGPB treatment, in both tissues.

**Table 4 metabolites-13-00374-t004:** Effect of *S. violaceoruber* culture seed-priming treatment on the germination index (%) and root and hypocotyl length of tomato seedlings.

Treatments *	Germination Index % ^#^	Root Lenght (cm) ^#^	Hypocotyl Lenght (cm) ^#^
T1	4.75 ᵃ ᵇ	1.11 ᵃ ᵇ	1.01 ᵃ ᵇ
T2	5.11 ᵃ	1.41 ᵃ	1.18 ᵃ
CTRL	4.61 ᵇ	1.06 ᵇ	0.89 ᵇ

***** Treatments were carried out for 7 days with three doses: 0 (CTRL) and 1:5 and 1:10 (T1 and T2, respectively) dilutions in distilled water of *S. violaceoruber* cultures. **^#^** Different letters indicate statistically significant differences (Tukey’s test, *p* < 0.05).

**Table 5 metabolites-13-00374-t005:** VOCs produced by *S. violaceoruber* identified using SPME-GC/MS.

t_R_ (min)	Compounds	Area %
7.75	Disulfide, dimethyl	76.89
20.09	Dimethyl trisulfide	1.09
25.60	Hexanoic acid, 2-ethyl-, methyl ester	14.74
36.41	2-Methylisoborneol	0.35
39.24	1*H*-Indene, 1-ethylideneoctahydro-7a-methyl	0.89
39.49	1*H*-Indene, 1-ethylideneoctahydro-7a-methyl-, isomer	0.41
51.52	Geosmin	4.02
53.69	Cadinene	1.61

**Table 6 metabolites-13-00374-t006:** VOCs produced by *S. lycopersicum* identified using SPME-GC/MS.

t_R_ (min)	Compounds	% Remaining on Treated Leaves vs. Ctrl
15.9	α-pinene	10.39
16.7	2-carene	10.93
17.4	α-terpinene	9.17
21.4	p-cymene	7.17
21.9	o-cymene	2.33
22.5	menthatriene	3.34
23.9	limonene	1.45
24.1	β-phellandrene	1.09
24.5	γ-terpinene	1.96

## Data Availability

Data supporting reported results can be found in this article or are available under request to corresponding Authors.

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
