# Peer review of "Bioactive Metabolite Survey of Actinobacteria Showing Plant Growth Promoting Traits to Develop Novel Biofertilizers"

_metabolites, 2023, doi:10.3390/metabo13030374_

Round 1

Reviewer 1 Report

The work presented by Faddetta et al. offers considerable insights into the use of actinobacteria to promote the development of Solanum lycopersicum. Overall, the work is really well structured and very clear, only the graphic representation part should be taken care of to enhance the results of the work. I invite the authors to add, if available, information on the ability of actinobacterial treatments to also promote the uptake of minerals essential for plant growth, since they are a limiting factor in plant growth. Below are just a few minor comments:

Line 168: 8L/m? check also line 169.

Figure 1: put Y axis in log scale.

Figure 3, 4, and 5 are of low quality and unattractive. 

Figure 4: put Y axis in log scale.

Author Response

Reviewer 1

add, if available, information on the ability of actinobacterial treatments to also promote the uptake of minerals essential for plant growth,

Authors

We thank the Reviewer for the suggestion. The required information has been emphasized in the revised version of the manuscript in introduction and discussion sections.

Reviewer 1

Line 168: 8L/m? check also line 169

Authors

The lines were corrected. We thank the Reviewer for the suggestion.

Reviewer 1

Figure 1: put Y axis in log scale.

Authors

Based on Reviewer comment, we used scientific notation in Y axis.

Reviewer 1

Figure 3, 4, and 5 are of low quality and unattractive

Authors

We are sorry and based on Reviewer comments we have changed figure 3 with a table (Table 4) and tried to improve the quality of figure 4 and 5.

Reviewer 1

Figure 4: put Y axis in log scale

Authors

Based on Reviewer suggestion we used scientific notation in Y axis

Reviewer 2 Report

Dear Authors

The present manuscript “Bioactive metabolite survey of actinobacteria showing plant growth-promoting traits to develop novel biofertilizers” demonstrated the three actinobacterial strains Streptomyces violaceoruber, Streptomyces coelicolor and Kocuria rizophilamolecular for their potential as plant growth promoters. The present study showed significant data for a better understanding of their interaction mechanism. Manuscript is very well organized and presented with nice discussion. Although there are some small queries, please find them below.

1.      Line 31- “global DNA methylation” please explain what does it mean?

2.      Line 175- The culture was used to treat the seeds before germination. The bacterial cultures were prepared in water for treatment or it was a suspension in growth media. If it is in growth media, did there a control treated with growth media only?

3.      Line 217- Morphological measurements of root and hypocotyl length were made by placing the 218 seedlings on graph paper and analyzed with ImageJ software. The biomass of seedlings were also tested?

4.      Line 421- Figure 5. Is there any significant difference?

Thank you

Author Response

Reviewer 2

Line 31- “global DNA methylation” please explain what does it mean?

Authors

According to Reviewer comment, an explanation has been provided in the text in Material and Methods section.

Reviewer 2

Line 175- The culture was used to treat the seeds before germination. The bacterial cultures were prepared in water for treatment or it was a suspension in growth media. If it is in growth media, did there a control treated with growth media only?

Authors

As reported in the manuscript, the bacterial cultivations were performed using R5A growth medium and then their PGP effects on tomato seeds have been tested using 1:10 dilution in distilled water of cultures (including cultivation broth and bacterial biomass). In our preliminary investigations we have observed that the use of R5A growth medium and dilutions thereof in distilled water have not any effect on S. lycopersicum seed germination and/or growth and development. Thus, from then on we have always used distilled water as control for the experiments. This point has been reported in the text of the revised manuscript in the discussion section.

Reviewer 2

line 217- Morphological measurements of root and hypocotyl length were made by placing the 218 seedlings on graph paper and analyzed with ImageJ software. The biomass of seedlings were also tested?

Authors

Thank you for this suggestion. We are sorry since, in this work, the biomass of tomato seedlings has not been measured because we were afraid that they were too young and small to appreciate any biomass differences. Anyhow, our focus was to investigate the PGP traits of the selected actinobacteria and, therefore, tomato seeds were used to perform biological assays. For this in vivo analysis, we decided to measure some parameters that we evaluated reliable and not too much labour intensive. Indeed, the measurement of a more exhaustive array of plant morphometric parameters - also including biomass as well as leaf size, chlorophyll content and so on - has been performed to investigate the PGP bacterial strain effect on adult plants that were cultivated in pots in phytotron . However, this are results that will be part of a different manuscript that we hope to publish as soon as possible.

Reviewer 2

Line 421- Figure 5. Is there any significant difference?

Authors

According to Reviewer 2 comments, the result of a one-way ANOVA test showing P< 0.05 has been indicated.

Reviewer 3 Report

The subject of the manuscript is interesting.

The subject of the manuscript is consistent with the scope of the Journal.

The abstract conveys the scope of investigations and conclusions drawn. The keywords correspond well to the scope of the research.

I think the paper needs some corrections:

1)  the research hypotheses have not been clearly formulated,

2) line 95 - Kieser et al. (2000)?, line 120 - Faddetta et al., 2021?, line 443 - Vessey (2003)?, lines 510-511 - Sudha et al. (2022)?, etc.;  the references should be cited as numbers,

3)  the Materials and Methods section requires some extension,

4) add information about analytical quality control for HPLC/MS/Q-TOF analysis,

5) add information about statistical analysis of results to Materials and Methods section,

6) add references for all analytical and statistical methods to Materials and Methods section,

7) correct grammatical errors, add necessary spaces or remove unnecessary spaces,

8) conclusions should be improved; conclusions must summarize the results obtained in experiment,

9) format References section according to Instructions for Authors,

10)  format all sections of the manuscript according to Instructions for Authors.

You must check your paper very exactly and correct all mistakes and complete lacking data of papers.

Author Response

Reviewer 3

the research hypotheses have not been clearly formulated,

Authors

According to Reviewer comments, the hypothesis has been clearly formulated in introduction section and emphasized on discussion section.

Reviewer 3

line 95 - Kieser et al. (2000)?, line 120 - Faddetta et al., 2021?, line 443 - Vessey (2003)?, lines 510-511 - Sudha et al. (2022)?, etc.;  the references should be cited as numbers

Authors

We thank the reviewer for this suggestion. Accordingly, text has been corrected.

Reviewer 3

the Materials and Methods section requires some extension

Authors

According to Reviewer criticism, Materials and Methods section has been modified to improve clarity

Reviewer 3

add information about analytical quality control for HPLC/MS/Q-TOF analysis,

add information about statistical analysis of results to Materials and Methods section,

add references for all analytical and statistical methods to Materials and Methods section

correct grammatical errors, add necessary spaces or remove unnecessary spaces,

conclusions should be improved; conclusions must summarize the results obtained in experiment,

Authors

We thank the reviewer for the comment and suggestions. Text has been changed accordingly.

Reviewer 3

format References section according to Instructions for Authors,

You must check your paper very exactly and correct all mistakes and complete lacking data of papers

Authors

Thank you for this suggestion. The Mendeley editor has been used for formatting References’ section and the lacking information has been added to references, accordingly.

Reviewer 3

format all sections of the manuscript according to Instructions for Authors

Authors

Thank you for this suggestion. We have used the MDPI template to format the manuscript and edited all sections.